# Enhanced hexamerization of insulin via assembly pathway rerouting revealed by single particle studies

Freja Bohr [1,2], Søren S. -R. Bohr [1,2], Narendra Kumar Mishra[1,5], Nicolás Sebastian González-Foutel[3,5], Henrik Dahl Pinholt[1,2,4], Shunliang Wu[1], Emilie Milan Nielsen[1,2], Min Zhang [1,2], Magnus Kjaergaard[3], Knud J. Jensen [1✉] & Nikos S. Hatzakis [1,2✉]

Insulin formulations with diverse oligomerization states are the hallmark of interventions for the treatment of diabetes. Here using single-molecule recordings we firstly reveal that insulin oligomerization can operate via monomeric additions and secondly quantify the existence, abundance and kinetic characterization of diverse insulin assembly and disassembly pathways involving addition of monomeric, dimeric or tetrameric insulin species. We propose and experimentally validate a model where the insulin self-assembly pathway is rerouted, favoring monomeric or oligomeric assembly, by solution concentration, additives and formulations. Combining our practically complete kinetic characterization with rate simulations, we calculate the abundance of each oligomeric species from nM to mM offering mechanistic insights and the relative abundance of all oligomeric forms at concentrations relevant both for secreted and administrated insulin. These reveal a high abundance of all oligomers and a significant fraction of hexamer resulting in practically halved bioavailable monomer concentration. In addition to providing fundamental new insights, the results and toolbox presented here can be universally applied, contributing to the development of optimal insulin formulations and the deciphering of oligomerization mechanisms for additional proteins.

[1] Department of Chemistry & Nanoscience Center, University of Copenhagen, Copenhagen, Denmark. [2] Novo Nordisk Foundation Center for Protein Research, Faculty of Health and Medical Sciences, University of Copenhagen, Copenhagen, Denmark. [3] Department of Molecular Biology and Genetics, The Danish Research Institute for Translational Neuroscience (DANDRITE), Nordic EMBL Partnership for Molecular Medicine, and Center for Proteins in Memory PROMEMO, Danish National Research Foundation, Aarhus University, Aarhus, Denmark. [4] Physics Department, Massachusetts Institute of Technology, Cambridge, MA 02139, USA. [5] These authors contributed equally: Narendra Kumar Mishra, Nicolás Sebastian González Foutel. ✉email: kjj@chem.ku.dk; hatzakis@chem.ku.dk

nsulin is a small protein produced by the β-cells of the pancreas that is crucial for regulating the blood glucose level in all animals[1]. Since 1922, insulin administration constitutes the hallmark of therapeutic intervention for diabetes and research has focused on the development of insulin analogs and formulations that act in either a rapid or protracted fashion[2–6]. Based primarily on ensemble recordings at the μM concentrations, the commonly accepted model for insulin self-assembly is a monomer-dimer-hexamer or monomer-dimer-tetramer-hexamer equilibrium[7–9]. These studies also showed the assembly process to be further stabilized by ions and excipients: The tetramer is stabilized by $Zn^{2+}$ and $Ca^{2+}$ via chelating to $His^{B10}$ and $Glu^{B13}$, respectively, while the hexamer is stabilized by the coordination of two $Zn^{2+}$ ions to $His^{B10}$ of three insulin dimers forming a toroidal hexamer[10–14]. The hexameric assembly is proposed to be further stabilized by phenolic ligands by inducing a conformational change from the T-state (tense) to the R-state (relaxed) forming the markedly more stable $T_3R_3$[15–21]. Understanding the abundance of each insulin oligomer and the mechanisms underlying the self-assembly properties of insulin and its analogs are essential for interpreting how native insulin is secreted from the pancreas and tailoring the properties of therapeutic insulin for fast or protracted action[4,22].

Insulin oligomerization studies have been impeded by experimental difficulties in directly observing all of the individual oligomers of the assembly process, challenges in reliable recordings at biologically relevant nM concentrations and discouraged by oversimplified models that fit these recordings. Ensemble methodologies correlate changes in a macroscopic property with the average oligomerization state (e.g., sedimentation equilibrium, stopped-flow, temperature jump kinetics, circular dichroism, and dynamic light scattering)[7,9]. Because bulk kinetics cannot directly measure the existence of all intermediates, researchers have relied on fitting the experimental observations with models intuitively assuming, albeit not directly observing, the addition of dimeric and tetrameric species, the first being consistent with structural evidence[10,23]. Assuming a three- or four-state equilibrium, reduces the number of unique transitions from 30, if all possible transitions occur, to 4 or 6 depending on the model used. This simplifies the analysis and allows extraction of equilibrium rate constants, but not the kinetic constants, from bulk data. The low sensitivity of these methodologies confines reliable readouts to the μM range, which is similar to the administrated insulin concentrations, but not the insulin concentration in the blood[24,25]. The considerable challenges these approaches impose are highlighted by the fact that, depending on the experimental condition and model, the calculated hexamerization constants $K_{MD}$ and $K_{DH}$ ($K_{MD}$: from monomer to dimer, $K_{DH}$: dimer to hexamer) vary by ~2 orders of magnitude, ranging from $K_{MD} = 10^3 \, M^{-1}$ to $K_{MD} = 10^5 \, M^{-1}$ and $K_{DH} = 10^8 \, M^{-2}$ to $K_{DH} = 10^9 \, M^{-2}$[7–9]. Precise knowledge of the effective concentration of each type of oligomeric species is key for the development of optimal insulin formulations. The existence of additional oligomeric forms[26] or oligomerization pathways involving monomeric additions would have a profound impact on the extracted average oligomeric form, association rates and equilibrium constants and their dependence on additives and insulin formulations; all of which are crucial for the development of optimal insulin formulations.

Here, using single-molecule studies, we directly observed the existence, abundance, and pathway organization of all intermediates of the self-assembly and disassembly process of insulin hexamers in equilibrium. We quantified the rate constants of association and dissociation that to the best of our knowledge has not been done before. While ensemble techniques require μM–mM insulin concentrations, similar to those found in formulations for pharmaceutical use, single-molecule techniques can directly observe phenomena in a concentration range similar to the pM physiological insulin concentration[27]. In this concentration range, our direct observations revealed previously unaccounted for monomeric additions occurring in all types of assemblies, thus prompting the revision of existing models that considered it to be negligible. The model-free analysis offered a comprehensive kinetic characterization of all possible pairs of monomeric, dimeric, and tetrameric assembly and disassembly and their dependence on excipients ($Zn^{2+}$ and phenol). Combined with rate simulations these data elucidated the relative abundance of each of the types of oligomer species across six orders of magnitude and their dependence on excipients. Our direct recording in the nM regime revealed a higher abundance of oligomers and subsequently a lower effective monomer concentration than previously reported[7]. We found hexamerization enhancement by additives to operate via re-routing the self-assembly pathway to favor dimeric or tetrameric addition.

## Results

**Direct observation of individual steps of the insulin self-assembly process.** We used Total Internal Reflection Fluorescence (TIRF) microscopy to directly observe the dynamic assembly and disassembly events of individual fluorescently labeled insulin monomers en route to hexamer formation. We chemically attached the fluorophore ATTO655 to $LysB^{28}$ on human insulin (in the following abbreviated $HI^{655}$) since it is well established not to interfere with insulin self-assembly (Fig. 1a, b)[28,29] (see Supplementary Fig. 1). In a typical experiment, 10 nM of $HI^{655}$ was allowed to equilibrate on a passivated microscopy surface (see Methods), resulting in its immobilization on the surface followed by stochastic binding and unbinding of species in solution. We acquired time series of TIRF images with hundreds of immobilized insulin molecules present in each field of view. Imaging with a low penetration depth allowed recordings of the dynamic assembly events on the microscopy surface with high signal-to-noise ratio, while particles in solution are not detected (see Fig. 1a). Using quantitative image analysis, we determined the x-y coordinates of each insulin particle with sub-pixel resolution[30–35] (Fig. 1c, d). We recorded time-dependent intensity fluctuations for each particle by integrating the intensity of each diffraction-limited spot[36]. The intensity changed in a stepwise manner (Fig. 1e, f, Supplementary Data 1) between several discrete levels, with a stochastically varying dwell time (residence time of each individual oligomeric species) and transition order. We, and others, have shown that such fluctuations directly correlate with the assembly and disassembly events for other fluorescently tagged proteins[30,33,37–39].

We performed several control experiments to establish that the distinct intensity shifts corresponded to binding and unbinding of insulin species and to confirm the validity of our readouts. High labeling efficiency and purity (Supplementary Figs. 2–10, Supplementary Scheme 1 and Supplementary Tables 1–6) minimized the potential bias of kinetics from unlabeled species. Fluorophore addition did not affect the kinetics or equilibrium[29] as shown by the recording of self-assembly for a 1:1 mixture of $HI^{655}$ and unmodified HI (Supplementary Fig. 11) as well as DLS measurements of unmodified HI and $HI^{655}$ (Supplementary Fig. 1). Similarly, fluorophore bleaching and blinking was quantified with immobilized monomeric biotin-labeled HI ($HI^{655}$-Biotin) (See Supplementary Figs. 12, 13 and Methods), which confirmed that fluorophore photophysics do not bias our readouts. Correction for the effects of fluorescent particles in solution, transient surface-docking, and fluctuations and unevenness of the laser excitation (Supplementary Fig. 14a–c) are described in Methods. Direct conversion of diffraction-limited

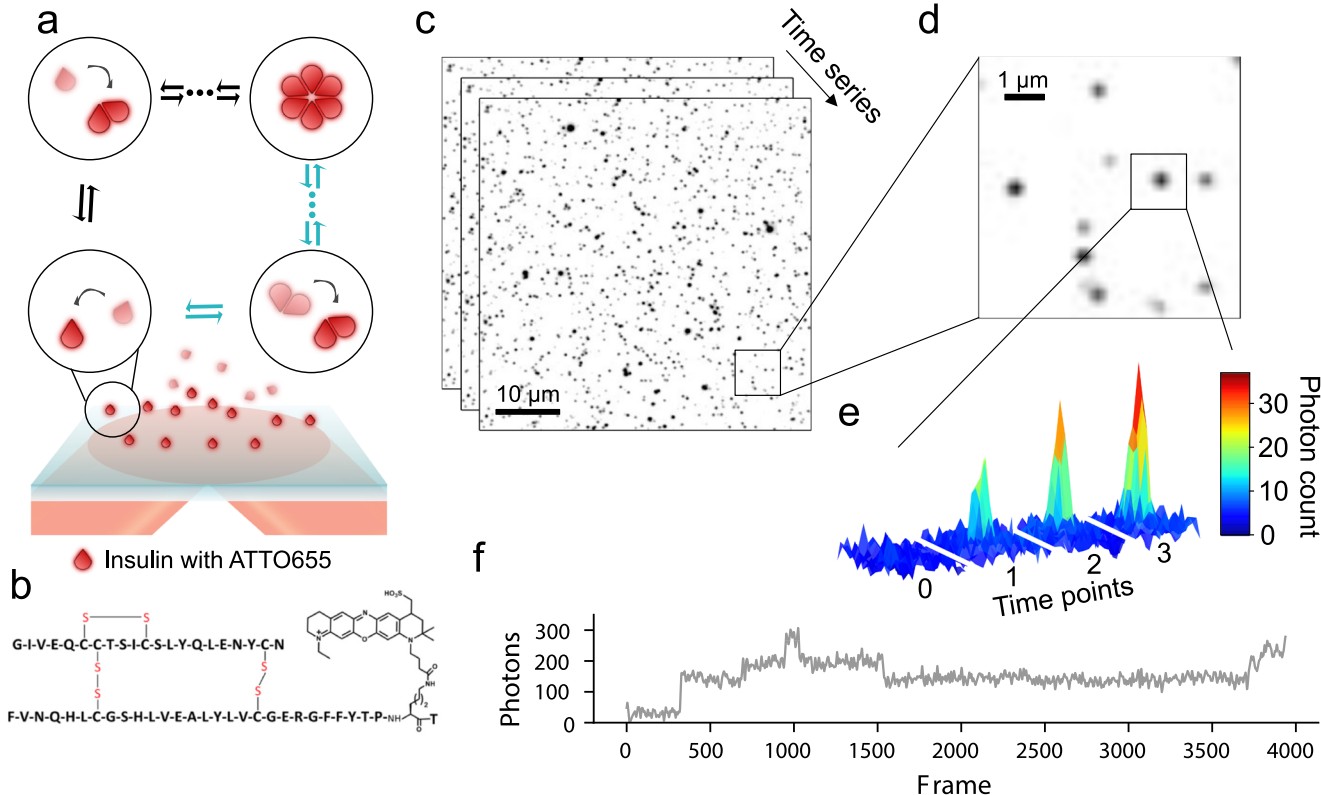

**Fig. 1 Experimental set up to observe oligomerization events of insulin using TIRF microscopy. a** Representation (not to scale) of the experimental setup. ATTO655 labeled Human Insulin (HI[655], see structure in **b**), are detected upon binding to the passivated microscopy surface for hexamerization. The evanescence field decays rapidly with the distance from the surface and ensures that the particles in solution are not detected (shaded red). This method allows for the direct observation of all types of oligomeric species addition. **c** Typical time series of micrographs recording the assembly of hundreds of surface-bound HI[655] in parallel (black spots) (scale bar 10 μm). **d** Close up, showing multiple insulin particles with varying intensity indicative of different oligomeric states (scale bar 1 μm). **e** The intensity of the spots resemble the point spread function (PSF) of a diffraction-limited spot suggesting they correspond to time-dependent intensity variations corresponding to single oligomerization events. **f** Typical single-molecule trajectory derived from **e** displaying discrete steps, the hallmark of single oligomerization events. Numerical Data for Fig. 1f can be found in Supplementary Data 1.

fluorescent readouts to photons (see Methods and Supplementary Fig. 14d–f) allowed interoperability across images and conditions[30,31,33,40] offering future comparisons to similar experiments with different experimental setups[41]. Consequently, these controls and calibrations ensured that the extracted intensity shifts correspond to binding and unbinding of insulin monomers and oligomers from the solution to immobilized insulin (see Fig. 1f, Supplementary Data 1 and Supplementary Fig. 15). We note that the reversible protein self-assembly events recorded here, are distinct from the irreversible, potentially toxic insulin aggregation[42,43]. We indeed have provided new super-resolution recordings revealing their abundance heterogeneous growth and kinetics[44,45]. The methodology allows the recordings of hundreds of individual insulin oligomerization processes and ~15,000 assembly or disassembly processes in a single experiment, providing the first real-time observation of individual insulin self-assembly events at the single-molecule level.

**Quantification of the abundance of oligomeric states and the kinetics of transitions between them**. The complexity of the self-assembly events is highlighted by the trace in Fig. 2a. At 10 nM, we found each trace to stochastically sample ~20 discrete transitions (Supplementary Figs. 15, 16). These events would be averaged out by bulk readouts, but are directly observed by the single-molecule recordings here. To classify the nature of the oligomeric species and extract the abundance and kinetics of all assembly and disassembly events, we used Hidden Markov Model

(HMM) analysis. We used a seven-state model representing the background and monomer through hexamer, which was fit to accurately describe transitions between all oligomeric states (see Methods for detailed description and Supplementary Figs. 17–20 and Supplementary Table 7). The horizontal red lines in Fig. 2a (Supplementary Data 2) represent a specific oligomeric state with a dwell time $\tau$ and vertical lines depict transitions between oligomeric states. The residuals between trajectory (gray) and HMM prediction (red) follow a Gaussian distribution around zero and thus indicate no systematic errors, confirming the accuracy of our seven-state model (Fig. 2a, bottom, Supplementary Data 2 and Supplementary Figs. 17c, 18, 19b, 20b, 21b). Calibration of photon to fluorophore ratio by immobilizing monomeric biotin-labeled HI (HI[655]-Biotin) revealed 46 photons per labeled monomeric insulin (Fig. 2b, Supplementary Data 3, see Methods for details and Supplementary Figs. 12, 13, 17). Our direct recordings here surprisingly revealed that the assembly and disassembly of insulin primarily operates via association and dissociation of monomers, and to a lesser extent via higher-order oligomers in the nM concentration range (Fig. 2c, Supplementary Data 4, and Supplementary Fig. 17). This hitherto unaccounted for monomeric assembly and disassembly are masked in bulk assays averaging the behavior of a large ensemble of unsynchronized molecules extensively discussed[7–10,12,20]. They are also reproduced here confirming that increasing concentration increases the average size, albeit not resolve individual oligomeric forms (see also Supplementary Fig. 1 and Supplementary Figs. 22, 23).

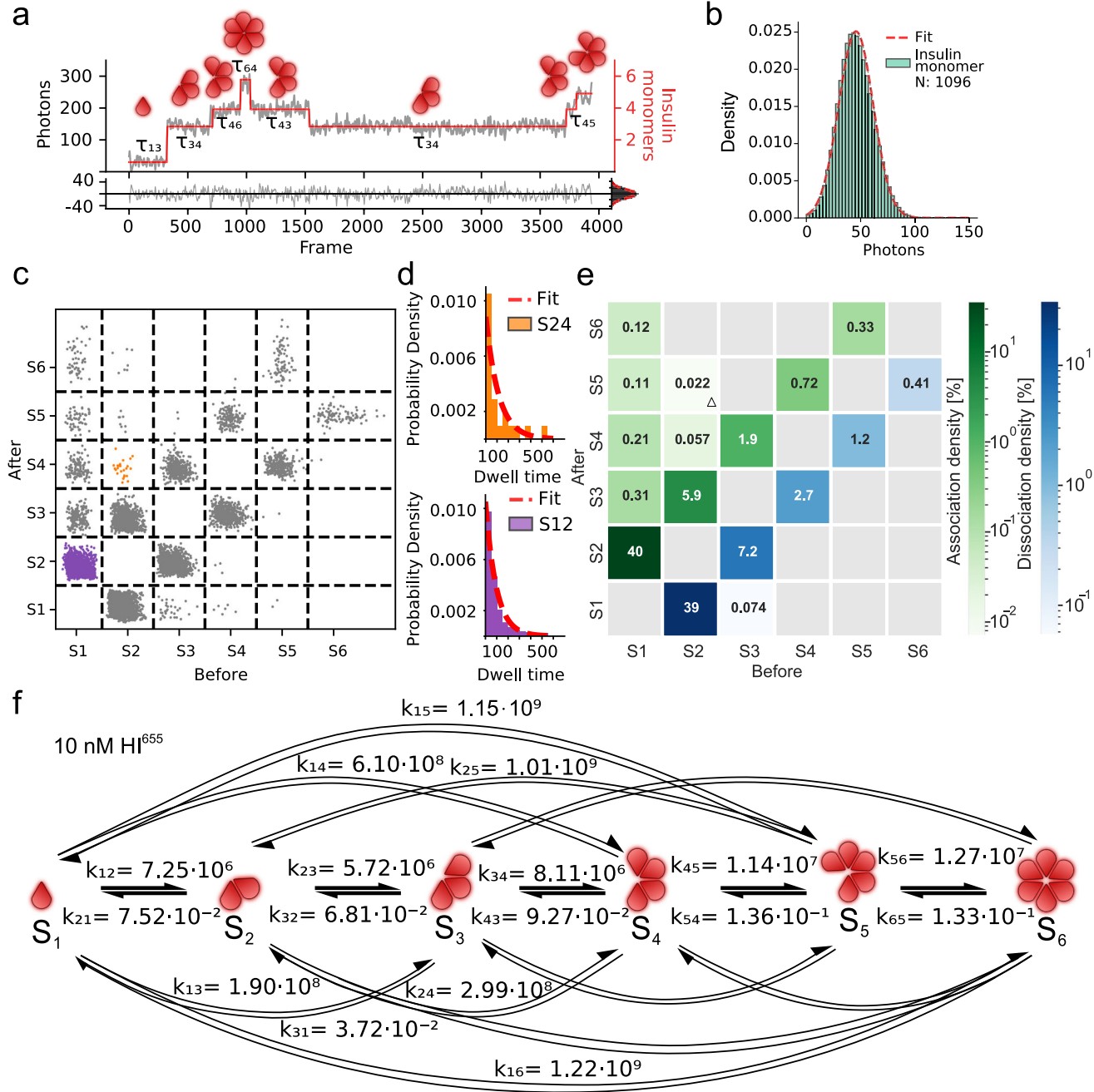

**Fig. 2 Single-molecule recordings of individual insulin hexamerization events allow for extraction of kinetic parameters. a** Representative trace showing discrete step-wise behavior (gray) corresponding to stochastic binding and unbinding of insulin during the hexamerization process in the absence of $Zn^{2+}$ and phenol. Hidden Markov Model (HMM) analysis using a seven-state model provides the idealized trajectory (red) and the extraction of the corresponding dwell times $\tau$ for each oligomeric state. Bottom: Residuals between the trajectory and idealized trajectory follow a normal distribution ($\mu = 0.0$, $\sigma = 14.0$) confirming unbiased fitting. **b** Monomeric $HI^{655}$ was imaged to calibrate the number of photons per monomer. A Gaussian fit (red line) of the intensity distribution of surface passivated $HI^{655}$ monomers revealed a mean photon value of 46 photons for a single insulin monomer ($\sigma = 16.0$) ($n_{videos} = 12$, $N_{particles} = 1096$). **c** Transition plot of idealized photon count found from HMM before and after a transition for 10 nM $HI^{655}$ ($n_{videos} = 4$, $N_{particles} = 2321$, $N_{transitions} = 48506$). Each cluster represents a specific transition separated by a grid (–black lines). The grid allows for the separation of transitions from one specific state to another. Purple denotes a transition from monomer to dimer (S1-S2), while orange is a transition from a dimer to a tetramer (S2-S4). **d** Dwell time distribution extracted from each cluster in **c** is fitted with a single exponential distribution (red dotted line) to obtain overall dwell time rate decay ($\tau$) (that is converted to rate constant ($k$)) and density for that specific transition. Data shown for S12 corresponds to a transition from monomer to dimer, ($N = 14501$), and S24 corresponds to a transition from dimer to tetramer, ($N = 21$). **e** CHESS (Complete HEatmap of State transitionS) plot, displaying the densities of association and dissociation occupancies for each pair of possible transitions. The densities indicate that oligomerization mainly happens via monomeric addition of insulin. The numbers within the squares correspond to transition densities. Gray squares are transitions with no data points. A triangle indicates rate constants calculated using less than 10 transitions and thus a large error. **f** Model-free extraction of rate constants for association [$M^{-1}s^{-1}$] and dissociation [$s^{-1}$] with HMM analysis for 10 nM $HI^{655}$ plotted for all observed oligomerization pathways. Notice the trend: the higher the association rate constant, the higher the dissociation constant for transitions involving higher-order oligomers. Numerical Data for Fig. 2a–d can be found in Supplementary Data 2, 3, 4, 5.

They however can have profound implications on existing models assuming dimeric additions as well as the effective insulin concentration in the blood.

To extend beyond the qualitative observation of monomeric assembly events, we determined the kinetic rate constants of each event. We used HMM to extract initial and final oligomeric state, their dwell times and transition densities as we have shown in the past[30,33,34] (Fig. 2c, Supplementary Data 4, and Supplementary Fig. 24). Each cluster of unique transitions contained up to ~15,000 transitions (see Fig. 2c, Supplementary Data 4, and Methods), which allowed us to extract the rates for each specific transition from the distribution of dwell times. This is shown for transitions from dimer to tetramer (Fig. 2c orange and Fig. 2d top, Supplementary Data 4, 5) and monomer to dimer (Fig. 2c purple and Fig. 2d, bottom, Supplementary Data 4–5, see Supplementary Figs. 25–28 and Supplementary Tables 8–11 for all transitions). For the dissociation processes, the unimolecular rate constant is directly extracted by the dwell time distribution. For the association processes, the corresponding biomolecular rate constant can be calculated by dividing by the solution concentration of the added species (see Methods). To estimate the solution concentration of each oligomeric species in these experimental conditions we fitted the histogram of all recorded assembly transitions with five Gaussian distributions (see Methods and Supplementary Figs. 17d, 18c, 19c, 20c, 21c). The association and dissociation rate constants and density for each unique transition from one state (x-axis) to another (y-axis) are summarized in the CHESS (Complete HEatmap of State transitionS) (Fig. 2e, see Methods for a detailed description). To the best of our knowledge this is the first direct and detailed extraction of rates constants for all intermediates of insulin hexamerization.

**Monomeric assembly and disassembly pathway for HI at nM concentration in the absence of additives.** The direct observation of self-assembly events combined with detailed, model-free analysis[46], allowed the extraction of the kinetic rate constants for 17 of the possible transitions involved in hexamer assembly and disassembly (Figs. 2f, 3a). Consistent with earlier studies, association rate constant were found to increase for transitions involving higher-order oligomeric states (e.g., $k_{34} = 8.1 \pm 0.3 \times 10^6 \, M^{-1} \, s^{-1}$ is 1.4-fold larger than $k_{23} = 5.7 \pm 0.1 \times 10^6 \, M^{-1} \, s^{-1}$) while the largest monomer addition rate constant observed for HI corresponds to a transition from pentamer to hexamer ($k_{56} = 1.3 \pm 0.1 \times 10^7 \, M^{-1} \, s^{-1}$, see Supplementary Table 12 for all data). Previously published monomer-to-dimer transition rates vary up to 20-fold depending on the experimental techniques used. The rate constants extracted here are a good match, though faster, to the literature values[7,8]. The single-particle sensitivity of the method allowed for the recording of monomeric additions (Fig. 2e and Supplementary Fig. 24) that would be averaged out by conventional assays focusing on dimeric additions (either monomer -> dimer -> hexamer or monomer -> dimer -> tetramer -> hexamer)[7,8,47]. Monomer assembly and disassembly represent the bulk of transitions observed (more than 99%, Fig. 2e), in contrast to previous studies that assume dimer assemblies and subassemblies (Supplementary Tables 12–15 and Supplementary Fig. 29b).

The high sensitivity of the TIRF assay allowed the recording of the often elusive dimer to tetramer transition and furthermore allowed for extracting the $k_{on}$ rate ($k_{24} = 3.0 \pm 0.6 \cdot 10^8 \, M^{-1} \, s^{-1}$). $k_{24}$ is 50-fold larger than $k_{23}$ ($k_{23} = 5.7 \pm 0.1 \times 10^6 \, M^{-1} \, s^{-1}$). Besides verifying the presence of dimers and hexamer in nM contraction it shows that the transition from dimer to a tetramer is kinetically favored over transition to trimers. This is consistent

with the assumption underlying assembly models that emphasize dimer addition. The transition from dimer to hexamer had too low an abundance (6 events out of 48,506 total transitions and 2321 trajectories, Fig. 4d, Supplementary Data 9, Supplementary Fig. 24a, and Supplementary Table 8) at this concentration. This supports HI oligomerization to mainly operate via monomeric addition, although the monomer-dimer-tetramer-hexamer equilibrium appears to be kinetically favored. Interestingly, while the community almost exclusively finds insulin monomers in nM concentration, we observe a diverse range of all oligomers and a relatively higher abundance of hexamers compared to literature (Fig. 2e, Supplementary Fig. 17 and Supplementary Table 7), drastically lowering the effective monomer concentration. This urges revision of current models on insulin hexamerization as well as improved calculations for effective concentration of each oligomeric form.

A crucial prediction from these rate constants is that the pathway of insulin assembly changes with concentration. We propose here a model where the self-assembly pathway of insulin is reliant on concentration: At nM concentrations similar to the secreted insulin concentrations used in the single-molecule assays, monomer addition dominates as a higher rate constant of dimer addition cannot compensate for the low population of dimers. At higher concentrations resembling pharmaceutical preparations and bulk experiments, the increasing dimer-to-monomer ratio shifts the assembly pathway towards dimer addition due to the intrinsically higher rate constant.

To test this model, we performed studies under otherwise identical conditions on the fast-acting NovoRapid insulin (Supplementary Figs. 21, 30), which has been altered to exhibit reduced dimerization[3,48,49]. Our size exclusion chromatography data showed the average size of NovoRapid to be less dependent on concentration as compared to HI and to remain smaller than HI in this concentration regime, supporting NovoRapid to primarily exist in monomeric form (see Supplementary Fig. 22 for SEC and Supplementary Figs. 1, 23 for DLS in agreement with SEC). While averaging techniques are unable to extract each individual intermediate the single particle results here revealed, the association rates involving dimer addition or higher-order oligomers and that they were reduced (e.g., $k_{24} = 2.0 \pm 0.3 \times 10^8 \, M^{-1} \, s^{-1}$ compared to $k_{24} = 3.0 \pm 0.6 \times 10^8 \, M^{-1} \, s^{-1}$ for HI[655]) supporting and extending current understanding. Interestingly, we found NovoRapid to display increased dissociation kinetics as compared to HI[655] (e.g., $k_{31} = 0.037 \pm 0.007 \, s^{-1}$ for HI[655], and $k_{31} = 0.062 \pm 0.009 \, s^{-1}$ for NovoRapid), suggesting NovoRapid has a different assembly and disassembly pathway than HI[655].

**$Zn^{2+}$ stabilizes the insulin hexamer and increases dimer additions.** To further evaluate our model, we used additives that are known to stabilize insulin hexamers in vivo such as $Zn^{2+}$ ions and the combination of $Zn^{2+}$ and phenol that are used in pharmaceutical formulations. $Zn^{2+}$ is expected to enhance the dimer-to-hexamer transition[11,12]. We quantified the effect of $Zn^{2+}$ by the addition of 100 µM $Zn^{2+}$, an excess amount of $Zn^{2+}$ in solution compared to HI. As expected, the addition of $Zn^{2+}$ resulted in an overall increase in association rate constants (see Figs. 3a, b, 4a, Supplementary Data 6, and Supplementary Figs. 18, 16, 29 and Supplementary Table 13). The rate constant for the dimer to tetramer transition increased 2.5-fold as compared to the $Zn^{2+}$ free condition (p-value = 0.009, $k_{24} = 3.0 \pm 0.6 \times 10^8 \, M^{-1} \, s^{-1}$ to $k_{24} = 1.0 \pm 0.4 \times 10^9 \, M^{-1} \, s^{-1}$), resulting in increased likelihood of forming a tetramer from dimer, rather than a trimer (See Fig. 3a, b and Fig. 4a, c, Supplementary Data 6, 8). Combined with the 1.2-fold increase in the

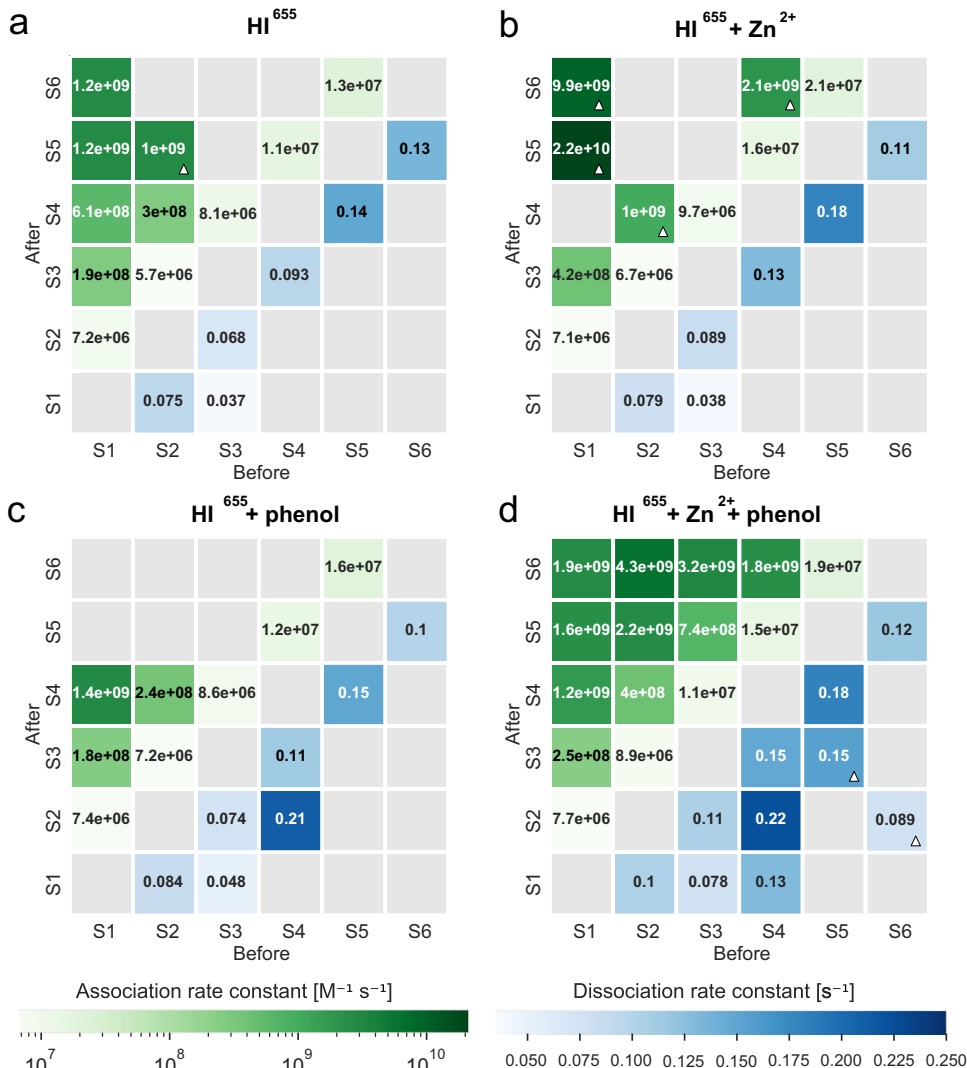

**Fig. 3 Additives stabilize the addition of dimeric and tetrameric species in insulin self-assembly.** CHESS plots displaying the extracted association [$M^{-1}\,s^{-1}$] and dissociation rate constants [$s^{-1}$] for each specific transition for the tested conditions: **a** 10 nM HI[655] ($n_{videos} = 4$, $N_{particles} = 2321$), **b** 10 nM HI[655] + 100 μM $Zn^{2+}$ ($n_{videos} = 4$, $N_{particles} = 2073$), **c** 10 nM HI[655] + 25 μM phenol ($n_{videos} = 7$, $N_{particles} = 3212$) and **d** 10 nM HI[655] + 100 μM $Zn^{2+}$ + 25 μM phenol ($n_{videos} = 5$, $N_{particles} = 2473$). Rate constants marked with white triangle are extracted from data with less than ten transitions. See Supplementary Tables 8–11 for the number of transitions in each square.

rate constant of monomer addition to trimers $k_{34}$ ($k_{34} = 8.1 \pm 0.3 \times 10^6\,M^{-1}\,s^{-1}$ to $k_{34} = 9.7 \pm 0.2 \cdot 10^6\,M^{-1}\,s^{-1}$) after the addition of $Zn^{2+}$, this results in an overall increase in tetramer density in the presence of $Zn^{2+}$ (see Fig. 4b, Supplementary Data 7, Supplementary Fig. 18a + 31a, b and Supplementary Table 7). The overall equilibrium shift to a hexamer is further compounded by the high transition rate from a tetramer to a hexamer ($k_{46} = 2.1 \pm 1.2 \times 10^9\,M^{-1}\,s^{-1}$), a transition that is practically not observed in the absence of $Zn^{2+}$ (see Fig. 3a, b, and Supplementary Tables 8–11). To substantiate this further we calculated the relevant equilibrium constants. $K_{56}$ increased from $0.96 \pm 0.01 \times 10^8\,M^{-1}$ to $1.95 \pm 0.15 \cdot 10^8\,M^{-1}$ ($p$ value $= 4.6 \times 10^{-5}$, see Fig. 4c, Supplementary Data 8). The overall reported earlier[7,16] 700-fold enhanced hexamerization of insulin by $Zn^{2+}$ appears to operate via accessing the rapid tetramer to hexamer transition. Similarly, the transition from dimer to hexamer is not observed (see Fig. 3b + 4d, Supplementary Data 9, and Supplementary Table 10) suggesting that in the presence of $Zn^{2+}$ insulin has an alternative favored hexamerization route that operates via monomer-dimer-tetramer-hexamer equilibrium.

**Phenol stabilizes the insulin hexamer and has no effect on dimer additions.** We then tested the effect of phenol, also known to stabilize the hexamer, via a conformational change from $T_6$ to the more stable $R_6$, and it is expected to do so without enhancing dimer association[50]. Indeed the addition of 25 μM phenol displayed no effect on dimer addition as compared to HI[655] ($p$ value $= 0.24$, $k_{24} = 3.0 \pm 0.6 \times 10^8\,M^{-1}\,s^{-1}$ to $k_{24} = 2.4 \pm 0.8 \times 10^8\,M^{-1}\,s^{-1}$) (Figs. 3a, c, 4c, Supplementary Data 8, Supplementary Fig. 27 and Supplementary Table 14). Similarly, the transition from tetramer to hexamer had a very low abundance (2 out of 60,176 transitions, Fig. 4d, Supplementary Data 9, and Supplementary Table 9). However, phenol shifted the equilibrium constant for the pentamer to hexamer transition by 55% to $K_{56} = 1.49 \pm 0.09 \times 10^8$ ($p$ value $= 1.4 \times 10^{-5}$, Fig. 3c + 4c, Supplementary Data 8). This appears to primarily originate from reducing the hexamer dissociation to pentamer ($k_{65} = 0.10 \pm 0.004\,s^{-1}$ with phenol compared to $k_{65} = 0.13 \pm 0.01\,s^{-1}$ without, $p$ value $= 0.0001$), supporting the stable formation of hexamer. Besides being consistent with the proposed role of phenol

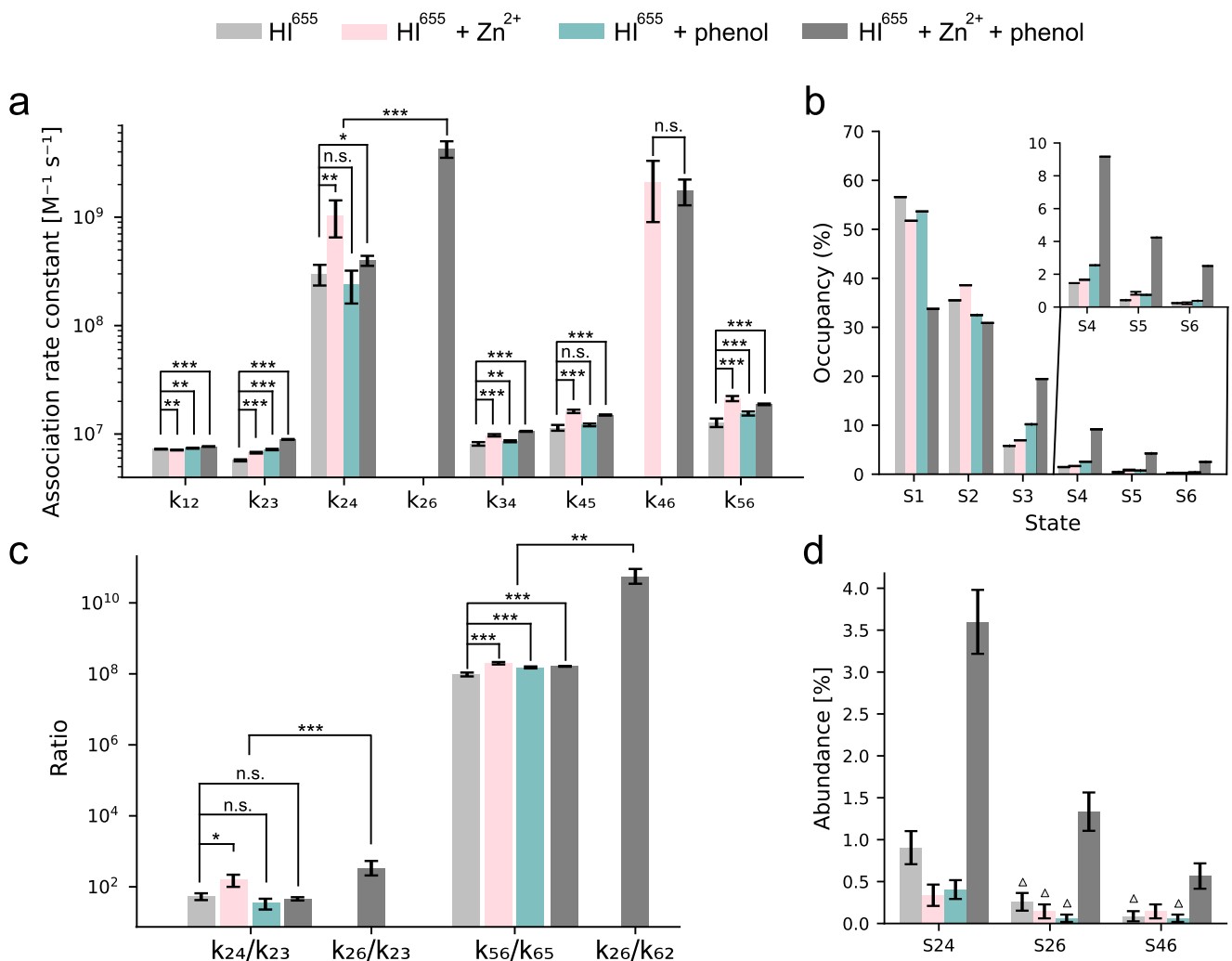

**Fig. 4 Insulin oligomerization pathway and its dependence on $Zn^{2+}$ and phenol. a** Rate constants for transitions involved in monomeric, dimeric, and tetrameric addition for all experimental conditions. The rate constant of the dimer to tetramer transition is two orders of magnitude faster than monomeric addition. **b** State occupancies for all conditions, showing that $Zn^{2+}$ and phenol increase the abundance of hexamers (see Methods). **c** Rate constant ratio between significant transitions. Rate constant ratio dimer/tetramer ($k_{24}$) or dimer/hexamer ($k_{26}$) and dimer/trimer transition ($k_{23}$). The increase in ratio suggests that the addition of $Zn^{2+}$ favors the addition of dimers via the tetramer over the addition of monomers, while phenol has no effect. No effect on $k_{24}/k_{23}$ is observed with both phenol and $Zn^{2+}$, suggesting that the hexamerization operates via a different route that involves the addition of tetrameric insulin species from solution, namely via $k_{26}$. The equilibrium constant ($K_{56} = k_{56}/k_{65}$) between pentamer and hexamer is increased by both phenol, $Zn^{2+}$ and a combination of the two. The effect of $Zn^{2+}$ and phenol is even bigger for $K_{26}$ ($k_{26}/k_{62}$) (not observed for other conditions). **d** Abundance of transitions related to the dimeric additions (S24, S26 and S46) for all conditions. Triangle denotes if the transition was observed too rarely to extract accurate rate constants. Errors are estimated as the square root of the number. $N_{S24} = (21, 7, 13, 89)$, $N_{S26} = (6, 3, 2, 33)$, $N_{S46} = (2, 3, 2, 14)$, $N_{particles} = (2321, 2078, 3212, 2473)$, $n_{videos} = (4, 4, 7, 5)$. Error bars for the rate constants are fit errors (of 4–7 measurements). The level of significance is determined by a Welch's t-test (*$p$ value < 0.05; **$p$ value < 0.01; ***$p$ value < 0.001; see SI/Methods for details). Numerical Data for Fig. 4a–d can be found in Supplementary Data 6, 7, 8, and 9.

stabilizing the hexamer by a conformational change from $T_6$ to $R_6$, these data further confirm the pathway rerouting we propose.

**Rerouting of the oligomerization pathway by the combination of $Zn^{2+}$ and phenol additives**. The combined effect of $Zn^{2+}$ and phenol offers a remarkable increase in the addition of dimeric and tetrameric species (Figs. 3d, 4c, Supplementary Data 8) as well as a significant equilibrium shift toward hexamer (Fig. 4b, Supplementary Data 7, Supplementary Fig. 31d and Supplementary Table 15). It results in 45-fold more favorable transition from a dimer to tetramer as compared to a transition to a trimer ($p$ value = $3.2 \times 10^{-8}$, $k_{23} = 8.9 \pm 0.07 \times 10^6$ $M^{-1}$ $s^{-1}$ to $k_{24} = 4.0 \pm 0.4 \times 10^8$ $M^{-1}$ $s^{-1}$, Fig. 4c, Supplementary Data 8). Similarly, the rate constant of hexamer formation directly from a tetramer is

120 times faster ($p$ value = $3.3 \times 10^{-5}$, $k_{45} = 1.5 \pm 0.01 \times 10^7$ $M^{-1}$ $s^{-1}$ to $k_{46} = 1.8 \pm 0.5 \times 10^9$ $M^{-1}$ $s^{-1}$). The high fidelity of the method allowed a direct observation of the transition from a dimer to a hexamer, a transition only sampled with enough statistics, when both additives are present (Fig. 4d, Supplementary Data 9, and Supplementary Fig. 28). The standard model of insulin hexamerization often accepts dimer to hexamer transitions to operate as a tri-molecular reaction involving three dimers[7,9]. Our measurements here also directly recorded a bimolecular reaction of the addition of tetrameric species to the insulin dimer on the surface, under the assumption that the concerted addition of two dimers does not occur rapidly compared to the 50 ms temporal resolution. The rate constant of transiting from dimer to hexamer is 480 times higher than

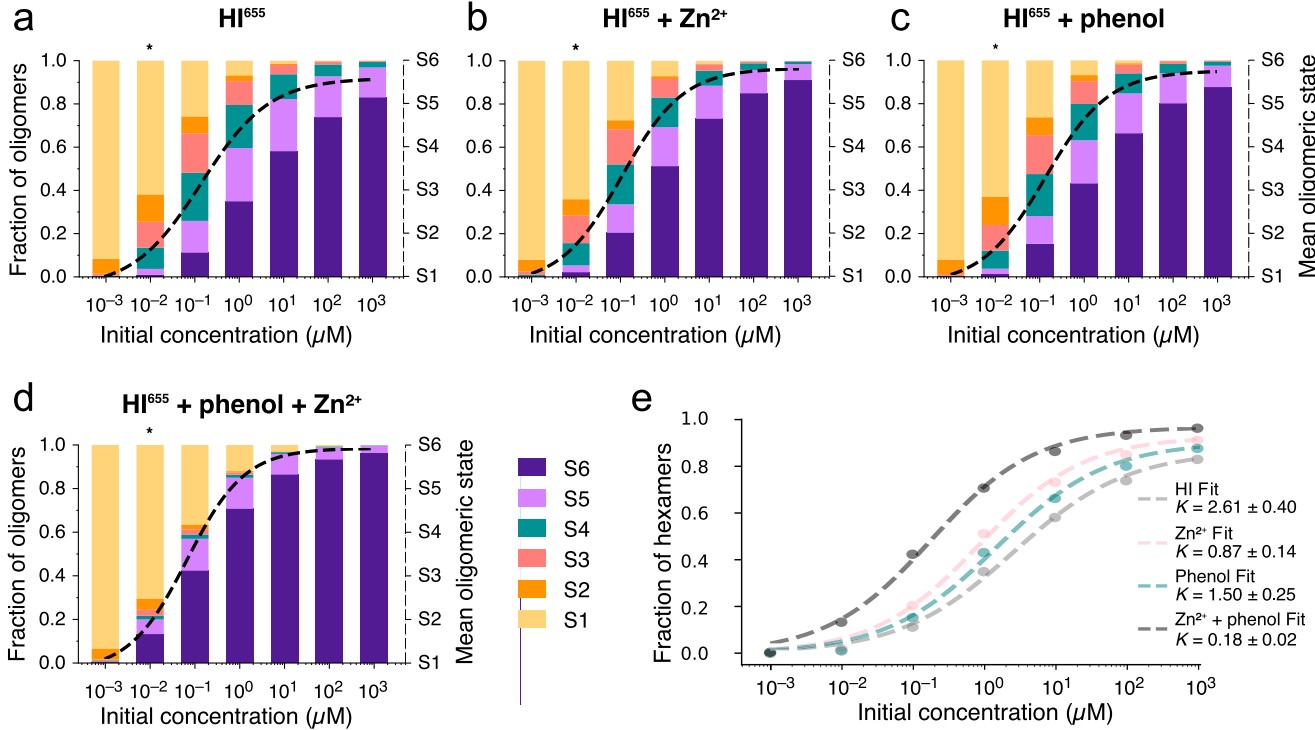

**Fig. 5 Extrapolation of the fraction of oligomerization species at varying initial insulin concentrations for each type of stabilizing additive. a–d** Stacked plots of the fraction of oligomeric species reached at numerically simulated equilibrium for different initial HI concentrations in solution (ranging from 1 nM to 1 mM) under the addition of different additives. Initial concentration represents the initial input monomer concentration before simulation has been started. (*) corresponds to the experimental conditions here of 10 nM HI[655]. The color code represents different oligomeric species. For each initial concentration, the mean oligomeric state (see Methods) is displayed as a dotted line, directly representing the ensemble averaging readout. The method allows deconvolution of all individual species. **e** Fit of the hexamer fraction (dark purple bar in **a–d**) for each experimental condition. See Methods and Supplementary Table 16 for the fitting details. Numerical Data for Fig. 5a–e can be found in Supplementary Data 10, 11.

transiting to a trimer ($p$ value $= 1.3 \times 10^{-6}$, $k_{26} = 4.3 \pm 0.7 \times 10^{9}\,\mathrm{M}^{-1}\,\mathrm{s}^{-1}$ to $k_{23} = 8.9 \pm 0.07 \times 10^{6}\,\mathrm{M}^{-1}\,\mathrm{s}^{-1}$, Figs. 3d, 4c, Supplementary Data 8). The calculated rate constant for a transition from a dimer to hexamer ($k_{26} = 4.3 \pm 0.7 \times 10^{9}\,\mathrm{M}^{-1}\,\mathrm{s}^{-1}$) and tetramer to hexamer ($k_{46} = 1.8 \pm 0.5 \cdot 10^{9}\,\mathrm{M}^{-1}\,\mathrm{s}^{-1}$) are similar (Fig. 3d and Supplementary Table 15) as one would expect to occur in solution, supporting that the surface immobilization is not biasing the readout. The equilibrium constant of tetrameric addition is $K_{26} = 4.8 \pm 2.1 \times 10^{10}\,\mathrm{M}^{-1}$, more than 100 times larger than the equilibrium constant $K_{56}$ (Fig. 4c, Supplementary Data 8, $p$ value $= 0.002$). The existing monomeric and dimeric pathways will have a small contribution to hexamerization. The overall increased hexameric form of insulin in the presence of both $Zn^{2+}$ and phenol appears to operate via selection of an alternative, faster oligomerization pathway, that of monomer-dimer-hexamer equilibrium.

**Bridging classical ensemble studies and single molecule approach.** The comprehensive extraction of multiple rate constants allowed us to simulate the time course of insulin assembly for initial concentrations varying by six orders of magnitude, from 1 nM to the mM range. This offered extraction of abundance of all oligomeric species in concentration ranges that are not always experimentally accessible with current published methodologies reporting the average behavior of a large ensemble of molecules, and we reproduced that here (Fig. 5 Supplementary Data 10–11, and Supplementary Figs. 1, 22, 23 and 32–35). We simulated the time evolution of each oligomeric species (S1–S6) using a reaction scheme with association and dissociation of monomers, dimers and tetramers (Supplementary Fig. 36), since these transitions are dominating the experimental observations.

The influence of $Zn^{2+}$ and phenol are not explicitly modeled but are implicitly considered in the rate constants. We simulated the four different initial conditions recorded in this study (HI[655], HI[655] + $Zn^{2+}$, HI[655] + phenol and the combination of $Zn^{2+}$ and phenol) using experimental rate constants as input (Supplementary Tables 17–20). In all simulations, kinetic equilibrium was reached within ~30 s, where the different oligomeric species stabilize at different levels depending on the presence of additives and the initial concentrations of HI (Supplementary Figs. 32–35). As our experiments do not consider the time evolution, we focused the analysis on the fraction of oligomers at the end point. Extrapolation of the fraction of oligomeric species for different concentrations displays the overall sigmoidal curve reported from modeling a monomer-dimer-(tetramer)-hexamer equilibrium[7], shifted to lower concentrations. In the absence of oligomer stabilizers, the predominant oligomeric species of insulin is the hexamer for initial concentrations above 1 μM. However, as unstabilized hexamers rapidly shed monomers and dimers, there are also considerable populations of tetramers and pentamers but also lower oligomers that are in general not accounted for by conventional analysis assuming equilibrium between three or four oligomeric species (Fig. 5a, Supplementary Data 10). Our results reveal a higher abundance of oligomers and hexamers in the nM-μM concentration range compared to literature that reports almost exclusively monomers and a small fraction of dimers, but no hexamers below μM[7]. In fact the abundance of hexamers for 10 nM initial concentration is 1% for HI[655] (Fig. 5a, Supplementary Data 10) and up to 13% with additives (Fig. 5d, Supplementary Data 10). This reduces the effective insulin monomer concentration by ~40% of what is currently expected[7] in the nM regime (Fig. 5a–d, Supplementary Data 10). Besides questioning

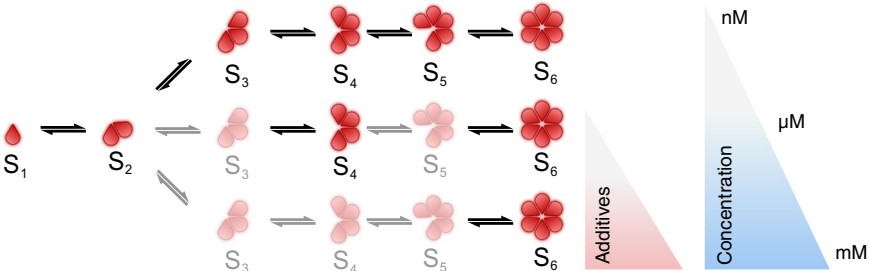

**Fig. 6 Model of pathway rerouting for insulin formulations by additives and its dependence on concentrations.** Understanding the mechanistic of pathway rerouting by additives and concentration enables emulation of the desired conditions or behavior that may be crucial for specific pharmaceutical targets.

current insulin assembly models, this reduction in effective insulin monomer concentration, would have profound implications in the amount of bioavailable subcutaneously administered insulin.

To compare with bulk experiments, we fitted the hexameric fraction increase with the Hill equation (Fig. 5e, Supplementary Data 11) for each experimental condition. This provides apparent affinity akin to what one would measure by bulk readouts and modeling with a three- or four-state equilibrium, (e.g., monomer-dimer-hexamer), but now derived from the high fidelity single-molecule recordings. The extracted Hill coefficient is consistent with reports on binding to multiple sites with different affinity (apparent Hill coefficient $n_h$ is <1)[51] (see Methods and Supplementary Table 16). In the presence of stabilizers, the transition from monomer to oligomers occurs at slightly lower HI concentrations reflected by changes in the apparent hexamer affinity (see Methods). The extracted apparent affinity is lower than the one calculated by bulk readouts. Consistent with earlier studies, the addition of $Zn^{2+}$ shifts the apparent affinity by ~3-fold (from 2.6 µM to 0.87 µM, while phenol has a more moderate effect resulting in 50% of the initial value. The combination of $Zn^{2+}$ and phenol has a remarkable reduction of 14-fold in the apparent affinity. Our results are, in general, consistent with previous studies that preferentially observed hexamers in the µM to mM range[50] and further hexamer stabilization by $Zn^{2+}$ and phenol but provide a mechanistic insight for this stabilization and self-assembly pathway rerouting.

## Discussion

The widely accepted model for insulin hexamerization is monomer-dimer-hexamer, while an additional tetrameric intermediate is sometimes also considered. In both cases, the hexamer formation is promoted by complexation with two $Zn^{2+}$ ions. These mechanisms are supported by bulk studies that correlate changes in a macroscopic property with the average oligomerization state and fitting the experimental observations with models assuming, albeit not directly observing, only dimeric and tetrameric additions. The relatively low sensitivity on the methods requires recording at the µM range, which is relevant to the administered insulin concentrations, but not the insulin concentration in the blood[24,25].

We directly observed insulin self-assembly events to be more complex than accounted for in any of the current models. We detect all oligomeric species between monomer and hexamers and observe stochastic transitions between them via monomeric and dimeric assembly and disassembly. This complexity imposes considerable challenges in current understanding, as it will be masked by conventional bulk readouts that average the behavior of a large ensemble of molecules. Indeed, to the best of our knowledge to date there is no experimental data, theory or simulation, supporting the existence of sequential assembly or

disassembly of monomeric insulin along the entire pathway. These results constitute the first direct validation that, contrary to current models, monomeric additions could occur to all types of oligomeric states, thus prompting a revision of existing models that intuitively assume insulin self-assembly to operate via the addition of dimers or tetramers.

The single-molecule recordings here transformed the stochastic nature of insulin assembly from an inaccessible problem in bulk assays to an experimental asset and offered the parallelized recording of the existence, abundance and dwell time of thousands of individual assembly and disassembly events of all types of insulin oligomeric species. This resulted in the model-free extraction of 17–25 kinetic rate constants for these transitions, which to the best of our knowledge has not been achieved before. The directly extracted rate constants here are faster than the rates calculated by previous studies based on fitting a three- or four-state equilibrium[7–9,52,53] and one cannot exclude that single-molecule readout on surface-immobilized molecules can have an effect. The wealth of control experiments compounded with the fact that they capture the proposed general trends, is consistent with no artifacts of the method. Notably, the rate constants of some transitions involving higher-order oligomers are close to what is considered the diffusion limit, where most collisions lead to binding. This is especially important for $k_{26}$ in the presence of $Zn^{2+}$ and phenol, further supporting that at sufficient concentrations a transition from a dimer to hexamer is the dominant mechanism.

We proposed here a model where the self-assembly pathway of insulin is rerouted by concentration, additives, and formulations (see Fig. 6 and Supplementary Fig. 37). At nM concentrations relevant for insulin secretion, monomer addition dominates. At higher concentrations resembling pharmaceutical preparations, the increasing dimer-to-monomer ratio shifts the assembly pathway toward dimer addition due to the intrinsically higher rate constant. Additives such as $Zn^{2+}$ and phenol may promote the addition of dimers. The combination of $Zn^{2+}$ and phenol, on the other hand, reroutes the hexamerization pathway, enhancing the direct transition from dimer to hexamer, a pathway rarely sampled by insulin alone. The fact that the presented method not only confirms the current theory of insulin oligomerization but also simultaneously captures each individual formation step with single-molecule resolution directly highlights the robustness of the analysis.

Insulin is believed to be secreted in hexameric form in vivo and to rapidly dissociate to monomers. Quantification of the dissociation of hexamers is therefore a key design parameter for pharmaceutical insulin formulations. The concentration range directly recorded here is relatively close to the insulin concentration under physiological conditions[27]. Our extraction of dissociation rates reveals dimeric dissociation to operate twice as fast as monomeric, a layer of information that is crucial for

understanding secreted insulin levels in blood and how the extent of oligomerization is regulated by $Zn^{2+}$ and phenol. Surprisingly, we found oligomers and hexamers present at nM concentrations relevant for subcutaneously administrated insulin as well as pancreatic insulin release, despite what current research states[7,24,25], effectively lowering the monomer concentration drastically with more than 40%. These findings together with the capacity of extrapolating behavior at higher concentrations for each type of formulation is crucially important for understanding and tailoring formulations for subcutaneous injection as well as understanding the metabolic processes of subcutaneously administered insulin.

These experimental observations support and augment existing knowledge, offering mechanistic insights into self-assembly pathway rerouting as a decisive element in enhanced hexamer formation by additives. Quantitative understanding of the processes and pathways that drive the association and dissociation of insulin and how they are remodeled by formulations and environmental conditions can aid both the optimized use of existing insulin formulations, the development of new novel formulations for optimized treatment of diabetes, and help guide the development of glucose-responsive insulins[54]. Besides deciphering the mechanism of insulin self-assembly regulation by additives, the work presented here establishes a universal methodological foundation for advancing our understanding of the regulation of the self-assembly process of additional biomolecular entities.

## Methods

**Materials**. All chemicals are of analytical grade and purchased from Sigma-Aldrich Denmark, unless otherwise stated. ATTO655-NHS ester was purchased from ATTO-TEC. Recombinant human insulin was purchased from Thermo Fisher USA. MilliQ water was used for aqueous preparations. The buffer used for all experiments was made from 10 mM $Na_2HPO_4$, 10 mM $NaH_2PO_4$, 5 mM NaCl, pH = 7.32.

### Insulin synthesis and labeling with chromophores (ATTO655)

*General*. High-resolution mass spectrometry was obtained on an UHPLC-MS with a QTOF Impact HD (Bruker) and Dionex UltiMate 3000 (Thermo) system equipped with a Kinetex® 2.6 μm EVO C18 100 Å column (50 × 2.1 mm, Phenomenex). Purification of conjugates was done on a Biotage-Isolera HPFC 300 system with a C18 column (SNAP Ultra, C18, 30 g). $CH_3CN{-}H_2O$ (0.1% HCOOH) was used as an eluent with a flow of 25 mL/min.

*Synthesis of ATTO655-human insulin*. Human insulin (21 mg, 0.0036 mmol, 3 equivalents) was suspended in 0.1 M tris buffer (0.2 mL), the pH was adjusted to 10.5 to dissolve it completely, ATTO655-NHS ester (1.0 mg, 0.00122 mmol, 1.0 equivalent) was dissolved in DMF (0.3 mL), added dropwise over 5 min to the stirring solution of Human insulin, and allowed the reaction mixture to stir for 15.0 min. The reaction was monitored by LCMS. Then the reaction mixture was diluted with 2.0 mL of $H_2O$ and pH was adjusted to pH 7.8. The product was isolated using RP-HPLC, on a Biotage SNAP ultra-column (C18, 30 g, 25 μm). $CH_3CN/H_2O$ mixed with 0.1% formic acid was used as eluents at a linear gradient of 5–50% $CH_3CN$ over 20 min, and a flow rate of 25 mL/min. Each fraction was analyzed through LCMS. Monosubstituted products were collected separately, $CH_3CN$ was removed at reduced pressure using rotary evaporator, followed by lyophilized to give product as a dark green (or blackish green) powder (HI655-Yield: 5.6 mg, 79%).

*Synthesis of Di-Fmoc-human insulin*. To the stirred solution of human insulin (300 mg, 0.0516 mmol) in tris buffer (100 mM, 4.0 mL) at pH 10.5, Fmoc-OSu (25 mg) dissolved in DMF (4.0 mL) was added, and the reaction mixture was stirred for 20 min at RT. LCMS analysis confirmed the completion of the reaction. The reaction mixture was diluted with water and pH of the reaction was adjusted from pH 10.5 to pH 7.8. The reaction mixture was purified on RP-HPFC (Isolera) using Biotage SNAP ultra-column C18, 60 g. The different fractions were analyzed by LCMS. The pure fractions were collected, concentrated, and then lyophilized to obtain the product *Gly*^A1 *Fmoc*-Lys^B29 *Fmoc*-HI as a solid fluffy white powder (*Gly*^A1 *Fmoc*-Lys^B29 *Fmoc*-HI, Yield 139 mg: 43%).

*Synthesis of Di-Fmoc-Phe*^B1 *-Biotin-PEG_3-HI*. Gly^A1 *Fmoc*-Lys^B29 *Fmoc*-HI (135.0 mg, 0.022 mmol) was dissolved in tris buffer at pH 10.5 (100 mM, 2.5 mL) and the pH of the solution reduced to 7.2. *Biotin-PEG_3-NHS-ester* (14.0 mg, 0.024 mmol) dissolved

in DMF (2.5 mL) was added portion wise over 5 min, the pH was further reduced to 7.0 and the reaction mixture stirred for 40 min at RT. The reaction was monitored by LCMS at defined time intervals, which confirmed completion of reaction after 40 min. Then, the reaction mixture was diluted with water (5.0 mL) and the pH was adjusted to 7.8. The reaction mixture was purified by RP-HPFC using Biotage SNAP ultra-column (C18, 60 g, 25 μm), using $CH_3CN/H_2O$ mixed with 0.1% formic acid with linear gradient of acetonitrile of 5–50%. Different fractions were separately analyzed by LCMS. Pure fractions were collected and freeze-dried to provide Gly^A1 *Fmoc*-Lys^B29 *Fmoc*-Phe^B1 *Biotin-PEG_3*-HI as a solid fluffy white powder (Yield 92 mg, 62.0%).

*Synthesis of Phe*^B1 *Biotin-PEG_3-HI*. Gly^A1 *Fmoc*-Lys^B29 *Fmoc*-Phe^B1 *Biotin-PEG3*-HI (90.0 mg, 0.013 mmol) was dissolved in DMSO (2.0 mL) over the stirring of 5 min at RT. 5% Piperidine in DMF (0.3 mL) was added to the stirred solution of Gly^A1 *Fmoc*-Lys^B29 *Fmoc*-Phe^B1 *Biotin-PEG_4*-HI and allowed the reaction mixture to stir for 10 min. Formation of product is confirmed by LCMS. Reaction mixture was further purified with RP-HPFC Isolera using Biotage SNAP ultra-column (C18, 60 g, 25 μm). $CH_3CN/H_2O$ mixed with 0.1% formic acid were used as eluents at a linear gradient of 5–60% $CH_3CN$ over 20 min, and a flow rate of 50 mL/min. Different fractions were separately analyzed through LCMS. Pure fractions were collected and freeze dried to get product *Phe*^B1 *Biotin-PEG_3*-HI as a solid fluffy white powder (Yield 60.0 mg, 72%).

*Synthesis of Phe*^B1 *Biotin-PEG_3-Lys*^B29 *Atto655-HI*. To the stirred solution of Phe^B1-*Biotin-PEG_3*-HI (25 mg, 0.0037 mmol) in tris buffer (100 mM, 1.5 mL) at pH 10.5, Atto-655-NHS-ester dissolved (1.2 mg, 0.0019 mmol) in DMF (1.5 mL) was added, and the reaction mixture was stirred for 15 min at RT. After 15 min, LCMS analysis confirmed that the reaction was completed. The reaction mixture was diluted with water and the pH was adjusted from 10.5 to 7.8. The reaction mixture was passed through a Biotage Snap BioC4 (300 Å, 10 g, C4, 20 μm) column with a flow rate of 12 mL/min with a gradient of 5–40% acetonitrile. Different fractions were separately analyzed by LCMS with a gradient of acetonitrile/water with 0.1% formic acid. Pure fractions were collected and freeze dried to provide Phe^B1-*Biotin-PEG3*-Lys^B29 *Atto655*-HI as a dark green fluffy solid as a product (4.0 mg, Yield 31%).

*V8 Enzymatic analysis of Phe*^B1 *Biotin-PEG_3-Lys*^B29 *Atto655-HI*. To verify the substitution pattern, Phe^B1-*Biotin-PEG_3*-Lys^B29 *Atto655*-HI was subjected to enzymatic digestion by treatment with endoproteinase Glu-C from Staphylococcus. Analysis of the fragments confirmed that Atto655 was covalently attached at Lys^B29 and Biotin-PEG_3 substituted at Phe^B1 position, respectively:

- Gly^A1-Glu^A4 ($C_{18}H_{32}N_4O_7$) Calculated: 416.22, Observed: 417.213
- Asn^A18-Asn^A21 + Ala^B14-Glu^B21 ($C_{59}H_{88}N_{14}O_{20}S_2$), Calculated [M + 2H]$^{2+}$: 689.29, Observed: 689.27
- Arg^B22-Lys^B29 *Atto-655*-Thr^B30 ($C_{82}H_{111}N_{16}O_{15}$), Calculated [M + 2H]$^{2+}$: 813.39, Observed [M + 2H]$^{2+}$: 813.37.
- Gln^A5-Glu^A17 + Phe^B1-*Biotin-PEG_3*-Glu^B13 ($C_{147}H_{229}N_{37}O_{48}S_5$) Calculated [M + 2H]$^{2+}$: 1722.271, Observed: 1722.25.

*Synthesis of Lys*^B29 *Atto655-insulin aspart*. Freshly purified insulin aspart (NovoRapid) (21 mg, 0.0036 mmol, 3 equivalents) was suspended in 0.1 M tris buffer (0.2 mL), and the pH was adjusted to 10.5 to dissolve it completely. ATTO-655-NHS ester (1.0 mg, 0.00122 mmol, 1.0 equivalent) was dissolved in DMF (0.3 mL) and added dropwise over 5 min to the stirred solution of insulin aspart, whereafter and allowed the reaction mixture to stir for 15.0 min. The reaction was monitored by LCMS[29,55,56]. Then the reaction mixture was diluted with $H_2O$ (2.0 mL) and the pH was adjusted to pH 7.8. The product was isolated using HPFC using a SNAP ultra-column (C18, 30 g, 25 um). $CH_3CN/H_2O$ mixed with 0.1% formic acid was used as eluents with a linear gradient of 5–50% $CH_3CN$ over 20 min, and a flow rate of 25 mL/min. Each fraction was analyzed by LCMS. Monosubstituted products were collected separately, $CH_3CN$ was removed at reduced pressure on a rotatory evaporator, followed by lyophilization to provide the desired product as a dark green (or blackish green) powder (Lys^B29 Atto-655-NovoRapid-HI-Yield: 5.6 mg, 79%).

*V8 enzymatic analysis of Lys*^B29 *Atto655-insulin aspart*. To verify the substitution pattern, Lys^B29 *Atto655-insulin aspart* was subjected to enzymatic digestion by treatment with endoproteinase Glu-C from Staphylococcus. Analysis of the fragments confirmed that Atto655 was covalently attached to Lys^B29:

- Gly^A1-Glu^A4 ($C_{18}H_{32}N_4O_7$) Calculated: 416.22, Observed: 417.213
- Asn^A18-Asn^A21 + Ala^B14-Glu^B21 ($C_{59}H_{88}N_{14}O_{20}S_2$), Calculated [M + 2H]$^{2+}$: 689.29, Observed: 689.27
- Arg^B22-Lys^B29 *Atto-655*-Thr^B30 ($C_{80}H_{107}N_{16}O_{20}S$), Calculated [M + 2H]$^{2+}$: 822.38, Observed: [M + 2H]$^{2+}$: 822.36.
- Gln^A5-Glu^A17 + Phe^B1-Glu^B13($C_{126}H_{197}N_{34}O_{41}S_4$), Calculated [M + 3H]$^{3+}$: 990.442, Observed: 990.422, Calculated [M + 4H]$^{4+}$: 743.084, Observed: 743.084.

*Sample preparation.* Degassed MilliQ $H_2O$ (1 mL) was added to HI[655], HI[655]-Biotin, and NovoRapid[655] (1.0 mg of dry powder) and shaken gently for 2–3 min, at which time it appeared as a suspension. The pH of the solution was increased to pH 10.0 by addition of 0.5 M NaOH (added 1–2 µL for 2–3 times to reach the pH of 10.0 to dissolve it completely as a transparent solution. Further pH was lowered with 0.5 M HCl and shaken gently so that cloudy insulin dissolved, and the solution appeared transparent. Afterwards, the pH of the solution was adjusted 7.4–7.5 using 0.2 M HCl. The solution was further filtered into another Eppendorf tube through 0.2 µm syringe filters in order to remove any precipitate/aggregates. The concentration of insulin was further determined on a Nanodrop instrument. The molar absorption coefficient value for Atto655 is $1.25 \times 10^5$ $M^{-1}$ $cm^{-1}$. After determination of concentration, stock solutions were utilized for further experiments.

**Dynamic light scattering.** DLS measurement was carried out on a Malvern Zetasizer (Malvern, United Kingdom) µV instrument at 25 °C using a 2 µl Quartz cuvette with 1.25 mm light path length. Hydrodynamic radius was calculated using a standard equation with dynamic viscosity of water at 25 °C which is embedded in Malvern program. Sample was measured at varying concentrations of 50 and 40 µM insulin in 10 mM $Na_2HPO_4$, 10 mM $Na_2HPO_4$, 10 mM $NaH_2PO4$, and 5 mM NaCl at pH 7.5. The mean hydrodynamic radius for each condition was found with a log-normal fit to the data.

**Size exclusion chromatography (SEC).** The samples (200 µL) were separated by size on a fast protein liquid chromatography system (GE ÄKTA Purifier 10 System with Monitor UV-900 and Sample Pump P-900). The SEC was carried on a calibrated Superdex 75 Increase 10/300 GL column at room temperature with a flow of 0.5 mL/min PBS buffer at pH 7.5 over 1.5 column volumes (CVs). The column was equilibrated over 2 CVs (1 CV = 24 mL) of running buffer before the sample injection with monitoring at 215, 280, and 663 nm. The molecular weight of each sample was calculated using a linear calibration curve of partition coefficient $K_{av}$

$$K_{av} = \left( \frac{retention\ volume - Void\ volume}{Column\ volume - Void\ volume} \right) \quad (1)$$

versus log(MW) generated by standard proteins; Conalbumin (75 kDa), Ovalbumin (43 kDa), Carbonic Anhydrase (29 kDa), Ribonuclear (13.7 kDa), and Aprotinin (6.5 kDa) (Fig. 1). The Stoke's radius ($R_s$) of each sample was calculated using a linear calibration curve of $\sqrt{-\log(K_{av})}$ versus $R_s$ of standard proteins; Conalbumin (36.4 Å), Ovalbumin (30.5 Å), Carbonic Anhydrase (23 Å), Ribonuclear (16.4 Å), and Aprotinin (13.5 Å) (Supplementary Fig. 22d).

**Microscopy surface preparation and surface passivation.** Microscope coverslips were cleaned thoroughly by sonication in 3 × 2% MilliQ Hellmanex solution, 3 × MilliQ and 1 × methanol for 15 min each. In between each round of sonication, the coverslips were rinsed with MilliQ. The clean coverslips were stored in methanol solution until usage. Clean coverslips were dried under a nitrogen flow and plasma cleaned for at least 4 min before attaching a sticky slide to it. Surfaces were passivated with 80 µL 100:1 mixture of 1 g/L PLL-PEG/PLL-PEG-biotin that incubated for at least 2 h. After incubation, the wells were thoroughly rinsed with MilliQ and functionalized with 80 µL 0.1 g/L neutravidin[30,39]. After functionalization the wells were again thoroughly rinsed and the surfaces were stored in MilliQ until insulin addition.

**Total internal reflection fluorescence microscopy (TIRFm) imaging.** Before imaging a video of an empty surface with only buffer was acquired for subsequent background correction. 350 µL insulin solution (10 nM HI[655]) in buffer was flushed into the chamber, and imaging was started immediately. All experiments were carried out in buffer at room temperature with $Zn^{2+}$ and phenol in excess amounts (100 and 25 µM). Data from at least four measurements were combined in all figures.

For HI[655]-Biotin control experiment, 350 µL 1 pM insulin was flushed into the chamber and immediately flushed extensively with buffer to remove any non-bound insulin. A total of 12 videos at different positions on the surface were recorded in a single chamber.

All experiments were conducted at a TIRF microscope (TIRFm, IX 83, Olympus) equipped with one EMCCD camera (Hamamatsu) and an oil immersion 100× objective (UAPON 100XOTIRF, Olympus). ATTO655 fluorophores were excited using a 640 nm solid-state laser line. Imaging was performed with 10% laser power (200 µW), 50 ms exposure time (followed 100 ms waiting time between frames resulting in a frame rate of 6.7 s$^{-1}$), 100 nm penetration depth and 300 EM gain. Imaging was done for a total of 4000 frames per video, resulting in a total imaging time of ~10 min.

**Image analysis.** Quantitative image analysis was performed using an in-house software[57] based on previous publications[30,31,58,59] and outlined in the following paragraphs. Traces were sorted based on different criteria, e.g., signal/background ratio, noisiness, and whether the traces displayed a clear stepwise behavior.

*EMCCD calibration.* Prior to the addition of particles, a control movie—with identical acquisition parameters to the subsequent measurement—was recorded on an empty sample for camera calibration (Supplementary Fig. 14d, Supplementary Fig. 14e shows a movie with particles for comparison). The calibration allowed conversion of intensity values into photons counts for subsequent measurements.

400 random pixels were selected from the control movie, forming 400 histograms of intensity values for each. Assuming a constant mean pixel value for such arrays, fluctuations are expected to arise only through EMCCD measurement noise. The noise is well described by the convolution of a Poisson distribution and an Erlang distribution. With the Poisson distribution modeling effects of shot noise and the Erlang distribution modeling the birth-death processes in the EM-gain. The resulting three-parameter distribution for pixel intensity is

$$p(s|s_0, \gamma, E) = \delta(s - s_0)e^{-E} + \sqrt{\frac{\gamma E}{s}} e^{-\gamma s - E} I_1(2\sqrt{\gamma Es}) \quad (2)$$

where $s$ is the pixel intensity and $I_1$ is the modified Bessel function of the first kind. The three parameters are $s_0$, a factory-set offset to the observed camera intensity, $\gamma$, the inverse gain of the camera, and $E$, the expected photon count from the pixel array[41,60,61]. Each of the 400 pixel arrays were fit with Eq. (2) using a chi-2 fit (Supplementary Fig. 14f). The average of all parameters with a *p*-value greater than 1% were then used to estimate the mean photon counts in pixels of subsequently acquired movies. This was done by simply offsetting and scaling the observed intensities

$$n = (s - <s_0>)<\gamma> \quad (3)$$

*Illumination profile correction.* To confidently compare photon counts across the images with fluorescent particles, a correction from the Gaussian illumination profile had to be performed (Supplementary Fig. 14a–c). The time-average of each movie was convolved with a Gaussian ($\sigma = 30$ pixels) to estimate the background illumination profile (Supplementary Fig. 14b). Each movie was then divided with its estimated background illumination profile to get the corrected set of movies (Supplementary Fig. 14c). Finally, to improve image contrast for subsequent particle identification, the movies were convolved with a Gaussian along the time axis ($\sigma = 3$ frames).

*Particle localization and signal extraction.* Localization of each particle was performed on an average representation of the time series since the particles are not moving. The x-y position of each fluorescent particle was determined by locating Gaussian-shaped peaks of fluorescence on the darker background using the python plugin Trackpy[30,31,58,59]. The specific parameters used for localizing particles in Trackpy were set as diameter = 11 pixels, sep = 6 pixels while minmass is calculated as *minmass = meanavgvideo · 0.4*, where meanavgvideo is the mean average pixel value of the Gaussian and illumination corrected video.

Subsequently, the exact position is refined using a 2D Gaussian fit to the PSF. The baseline (called *b*) for the 2D Gaussian fit was used for background correction. After localization, the signal was integrated with a roi (region of interest) of 9 pixels in diameter in all 4000 frames and lastly corrected for local background variations by subtracting the background value found from Gaussian fitting to obtain a single background corrected trajectory as:

$$f = \sum_{N_{pixels}} s - b * N_{pixels} \quad (4)$$

where $s$ is the pixel intensity, $b$ is the baseline from 2D Gaussian fit and $N_{pixels}$ represents the number of pixels within a roi (region of interest).

*Intensity calibration to number of insulin monomers.* Photon to fluorophore calibration allowed conversion of intensity shifts to discrete oligomeric state assembly and disassembly. Visual inspection of the trajectory displayed in Fig. 2a (Supplementary Data 2) suggests that insulin assembly and disassembly events primarily result in a signal change in integers of ~50 photons. To validate this, we directly converted the diffraction-limited fluorescent readouts to photons from 12 experiments with a surface-passivated biotinylated monomeric insulin (HI[655]-Biotin, $N = 1096$ particles) (Fig. 2b, Supplementary Data 3). Trajectories arising from aggregates were discarded. The methodology allows interoperability across experimental conditions and setups[41]. Analysis revealed an excellent agreement between both visual inspection, model prediction and Gaussian fit ($\mu = 46$ photons, $\sigma = 16$ photons) to monomeric insulin (Supplementary Fig. 12a, b). We excluded artifactual monomeric addition events by ensuring minimal bias from the surrounding via a *roundness* parameter allowing us to exclude incidents where particles were localized in the same pixel. The width of the distribution is similar to the width of the residual (Fig. 2a, bottom, Supplementary Data 2) further supporting the validity of the assay ($\sigma = 14.0$ compared to $\sigma = 16.0$, Fig. 2b, Supplementary Data 3).

Similar analysis was performed under addition of phenol, to ensure minimum bias from phenol on fluorescent readout ($N = 781$). As expected, no effect was found ($\mu = 41$ photons, $\sigma = 14$ photons) (Supplementary Fig. 13a).

**Bleaching and blinking control experiment.** Experiments with HI[655]-Biotin were used to quantify bleaching and blinking. Trajectories arising from aggregates were discarded. Bleaching analysis was made in increments of 400 frames. If the mean photon count for a specific particle was equal to or less than 30 photons in an

**Table 1 Hidden Markov Model input as initial guesses.**

| Distribution: | S0 | S1 | S2 | S3 | S4 | S5 | S6 |
|---|---|---|---|---|---|---|---|
| Input [photons]: | 20 ± 25 | 50 ± 25 | 100 ± 35 | 150 ± 35 | 200 ± 35 | 250 ± 35 | 300 ± 35 |

increment, it was denoted as bleached. Blinking analysis was performed on trajectories from frame 100 to frame 1000 to remove any bias from bleaching. If the residual (photon count in a specific frame) was bigger than or equal to 3 standard deviations of the trajectory it was denoted as a dark state, i.e., the fluorophore is blinking (Supplementary Fig. 12c–e). Similar analysis was performed under addition of phenol, to ensure minimum bias from phenol on fluorescent readout. As expected, no effect was found (Supplementary Fig. 13c).

**Insulin oligomerization experiment.** Experiments with HI[655] nonspecifically binding to the surface were used to observe and quantify single oligomerization events. The traces for the same conditions (insulin-, Zn[2+] and phenol concentration) from different surfaces were merged for later use. The summed photon histogram for all trajectories for 10 nM HI[655] followed a Gaussian mixture model of seven Gaussian consistent with a seven-state model accounting for the six steps of hexamerization plus background (essentially no particles observed) (Supplementary Fig. 17a). Thorough investigation into the fit revealed that the best seven distributions were found to be as listed in Table 1 and were applied to the HMM analysis. A hard threshold for photon count was set to be 500 photons, and traces containing higher photon count than this were discarded before the HMM analysis as they were higher-order aggregates.

For 10 nM NovoRapid[655] the signal to noise was lower than for experiments with HI[655]. For this reason, we used Gaussian smoothing with a width of $\sigma = 5$, on the raw trajectories for 10 nM NovoRapid[655] to obtain comparable signal to noise (Supplementary Fig. 17c and Supplementary Fig. 21b).

**Hidden Markov Model analysis.** The HMM data analysis was implemented using the Pomegranate package for Python similar to published methodologies[30,33,62]. Due to the low occupancy of the higher order oligomers of the insulin (tetramer, pentamer, and hexamer, Supplementary Table 7), the state values had to be frozen in order for the HMM to fit the entire dataset. For each trace, a model with 7 states (as defined in Table 1) was fitted. The idealized photon histogram was fitted with a mixture of seven Gaussians to extract state occupancies (Supplementary Figs. 17b, 18a, 19a, 20a, 21a). To determine the error, we employed a parametric bootstrapping approach, where the fit was reinitiated ten times to generate a bootstrap distribution of fit parameters estimates. From each parameter distribution, the mean and error were determined.

Although fluorescent intensity values are not technically Gaussian distributed, it has in practice been shown to be a robust method with little discrepancy when the values are distributed far away from 0 due to central limit theorem. Also, the HMM fit was evaluated by plotting the residuals that showed no systematic error of HMM fit for all conditions (Supplementary Figs. 19c, 18b, 19b, 20b, 21b).

**Determination of transition rate constants.** The idealized traces were further investigated by plotting all transitions in a Transition Density Plot as this is the current established method. Extraction of specific transitions is classically performed using fitting (e.g., k means clustering[30,33]) for simpler systems with fewer transitions, to reliably identify up to 64 clusters in total for all conditions. However, this method was not optimal for this data due to the many transitions possible and low sampling of higher-order transitions/states. This is despite very thorough examination of the data and extensive optimization of initial guesses for x-y positions of each cluster.

Here we applied a series of thresholds that can be subjected to extensive analysis to extract kinetic and thermodynamic insights. The grid thresholds were made in agreement with the HMM input, so that a specific state or transition found by the HMM, would be correctly separated by the grid. This allowed for a total theoretical number of 42 separable clusters (excluding those involved with S0). The TDP plots showcase insulin association and dissociation at 10 nM concentration regime to operate primarily from monomeric additions and to lesser extent via dimer or higher order oligomer addition (Supplementary Fig. 24 and Supplementary Tables 8–11).

Transition rate constants were found by fitting the dwell times contained in every cluster with a single exponential decay using a maximum likelihood fitting scheme (Supplementary Figs. 25–28 and Supplementary Fig. 30). Dwell times above 75 s were not fitted to avoid long-lived outliers that do not follow a single exponential decay (less than 5% of observable dwell times)[63]. The occupancy for each transition represents how many times the transition was observed.

Because dissociation of oligomers is unimolecular and association a bi-molecular reaction and thus oligomer concentration dependent, the decay rate for association was converted to rate constant by dividing with oligomer concentration in solution. Solution concentration can ideally be measured by the density of each oligomer landing directly on the surface. In order to get statistical significance, we estimated the oligomer concentration in solution by fitting the histogram of assembly transitions with five Gaussian distributions (representing monomer, dimer, trimer, tetramer and pentamer addition). In short, the histogram revealed 5 distributions corresponding to monomer, dimer, trimer, tetramer, and pentamer addition, respectively, and the weight from each distribution denotes the percentage of free soluble oligomers (See Supplementary Figs. 17d, 18c, 19c, 20c, 21c).

The found rate constant for association and dissociation can be found in Supplementary Tables 12–15, Supplementary Figs. 25–28 and Supplementary Fig. 30).

**CHESS plot.** We constructed CHESS (Complete HEatmap of State transitionS) for a transparent and convenient way of inspecting the multidimensional kinetic and thermodynamic information (Fig. 2e, Fig. 3, Supplementary Fig. 31). The x-axis shows the state before a transition, while the y-axis shows the state after a transition. Each square denotes one transition, the kinetic (rate constant) and/or thermodynamic (transition density) parameter written as the number in the square. The color code of each square represents the value written inside. Shown in Fig. 2e with transition densities for association and dissociation for all quantified transitions for 10 nM HI[655].

**Calculation of free energies.** Free energies were calculated based on transition state theory, where the monomer state was considered a ground state with relative energy 0 kJ/mol[63]. The relative free energy difference between states are

$$\Delta G = -RT\ln(K_{eq}) \qquad (5)$$

where $R$ is the gas constant and $T = 298$ K. Equilibrium constant $K_{eq}$ is given by

$$K_{eq} = \frac{k_{ij}}{k_{ji}} \qquad (6)$$

where $k_{ij}$ and $k_{ji}$ are transition rates between two states $S_i$ and $S_j$. The Gibbs energy of activation (energy barrier) for each transition is given as:

$$\Delta G^{\ddagger} = -RT\ln\left(\frac{hk_{ij}}{k_BT}\right) \qquad (7)$$

**Statistics and reproducibilty.** All values reported are the average at minimum of four independent replicates from separate experiments. Error bars represent SD as indicated in the corresponding figure legends, unless otherwise stated.

Level of significance is determined by a Welch's t-test (*p value < 0.05; **p value < 0.01; ***p value < 0.001) if not stated otherwise in the text.

**Simulations.** Simulations of the time course evolution of oligomeric species concentration were performed using KinTek Global Kinetic Explorer software[64] based on a kinetic model where the oligomeric species (S2 to S6) resulted from monomer and dimer association and dissociation steps as described in Supplementary Fig. 36a. As input for the simulations, we used experimentally obtained association rate constants ($k_{12}$, $k_{23}$, $k_{34}$, $k_{45}$, $k_{56}$, $k_{24}$, $k_{26}$ in $\mu M^{-1} s^{-1}$) and dissociation rate constants ($k_{21}$, $k_{32}$, $k_{42}$, $k_{43}$, $k_{54}$, $k_{62}$, $k_{65}$ in $s^{-1}$) for each of the four experimental setups (Supplementary Tables 17–20). The only constant missing from all experimental data sets is dimer dissociation rate, $k_{64}$. This value was too rare to be quantified, which suggests that the value is low in all cases. We can thus assign an upper bound based on the number of events that would be observable. For $k_{64}$, we used a value of 0.01 $s^{-1}$ which is an order of magnitude slower than the monomer dissociation from the hexamer. The rate $k_{42}$ could only be reliably extracted for the experiments containing phenol and $Zn^{2+}$ + phenol and the experimental value from phenol was extended to the other conditions (HI and HI + $Zn^{2+}$). The missing dissociation rate constants were supplied as follows: $k_{64}$ was arbitrarily set to an upper limit value of 0.01 $s^{-1}$ for the four conditions. $k_{42}$, obtained from the experiment with phenol, was used to perform simulations in the presence of $Zn^{2+}$ and in absence of oligomer stabilizers. $k_{26}$ and $k_{62}$ were only observed for the combination of $Zn^{2+}$ and phenol, and were set as zero in other conditions. The reactions were simulated using a default sigma value of 0.001 in 200 steps for a reaction time of 300 s with HI[655] (S1) initial concentration varying from 10 nM to 100 mM. The oligomer fractions were then calculated considering 1 the sum of the endpoint concentrations of each oligomeric species (Supplementary Figs. 32–35).

The mean oligomeric state was calculated as

$$\mu = 1 \cdot S1 + 2 \cdot S2 + 3 \cdot S3 + 4 \cdot S4 + 5 \cdot S5 + 6 \cdot S6 \qquad (8)$$

where S1–S6 represent the fractions of monomers to hexamers displayed in bars in Fig. 5A–D.

Evolution of hexamers in Fig. 5A–D (Supplementary Data 10) have been fitted with the Hill equation to find apparent affinity $K$:

$$f(x) = \frac{B_{max} \cdot x^{n_h}}{K^{n_h} + x^{n_h}} \tag{9}$$

where $K$ is the concentration needed for half-maximum hexamer formation. $n_h$ is the hill coefficient, and $B_{max}$ is the maximum hexamer fraction[51,65].

**Reporting summary**. Further information on research design is available in the Nature Portfolio Reporting Summary linked to this article.

## Data availability

The authors declare that the data supporting the findings of this study are available within the paper and its Supplementary information files. Source data behind Figs. 1–5 are available as Supplementary Data 1–11. Additional and relevant data are available from the corresponding authors on reasonable request. All data are available at ERDA repository of University of Copenhagen at https://sid.erda.dk/sharelink/dc8HiWatpL[66].

## Code availability

Codes used for localization, extraction of single-molecule trajectories, and HMM fitting can be found at https://doi.org/10.5281/zenodo.7341165.

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

## Acknowledgements

N.S.H. and K.J.J. acknowledge funding from Villum foundation center BIONEC (grant 18333). N.S.H. acknowledges funding from Villum foundation project (grant 40801), Carlsberg Foundation Distinguished Associate Professor Program (CF16-0797), and the NovoNordisk Center for Biopharmaceuticals and Biobarriers in Drug Delivery (NNF16OC0021948). The Novo Nordisk Foundation Center for Protein Research (CPR) is funded by a generous donation from the Novo Nordisk Foundation (grant no. NNF14CC0001). N.S.H. is a member of the Integrative Structural Biology Cluster (ISBUC) at the University of Copenhagen.

## Author contributions

N.S.H. designed the study with K.J.J. with serious inputs from F.B. and S.S.-R.B. N.K.M. performed insulin synthesis and labeling, purification, characterization, enzymatic assay, and dynamic light scattering experiments. F.B. performed all TIRF microscopy experiments and designed the controls, together with E.M.N. and M.Z. S.S.-R.B. and F.B. developed the data analysis, and H.D.P. developed the method for conversion from intensity to photons. F.B. carried out all data analysis and interpretation. N.S.G.F. and M.K. performed simulations. N.S.H. in tight interaction with K.J.J. had the overall supervision of the project. F.B. and N.S.H. wrote the manuscript with input from all co-authors. Freja Bohr: bohr@chem.ku.dk. Søren S.-R. Bohr: soeren@chem.ku.dk. Narendra Kumar Mishra: narendra@chem.ku.dk. Nicolás Sebastian González Foutel: nfoutel@mbg.au.dk. Henrik Dahl Pinholt: pinholt@mit.edu. Shunliang Wu: sw@chem.ku.dk. Emilie Milan Nielsen: emilieemnielsen@gmail.com. Min Zhang: min.zhang@chem.ku.dk. Magnus Kjaergaard: magnus@mbg.au.dk. Knud J. Jensen: kjj@chem.ku.dk. Nikos S. Hatzakis: hatzakis@chem.ku.dk.

## Competing interests

The authors declare no competing interests.
