## [Peer Review File · Communications Biology]

Reviewers' comments:

Reviewer #1 (Remarks to the Author):

In this article Bohr et al have utilized single molecule imaging to characterize the oligomerization behaviour of atto655 labelled human insulin (HI655). The authors have shown that labeling of HI by atto655 does not alter the oligomerization equilibrium. Hence, it is possibly safe to assume that oligomerization of HI655 is quite similar to that of unlabeled HI. They have used TIRFM to the record photon count (time) traces of single molecules of monomers and oligomers of HI655. The oligomeric status is determined from the magnitude of the photon counts per second. Recording of time trace has allowed the authors to follow the dwell time of each oligomeric species and the transition events of each oligomer to another type of oligomer. Use of wide field detection allows recording of thousands of such events enabling them to extract the equilibrium and the kinetic parameters of the oligomerization process. One might assume that labeling of HI by atto655 could affect its oligomerization properties. However, the authors have verified that HI655 behaves similarly as HI. One might also think that HI immobilized on the glass would behave differently from the HI in the solution. However, authors have conducted ensemble experiments to verify that the results are consistent with the equilibrium constants obtained from the single molecule experiments. Typically, analysis of the photon count traces involving six different species (monomers to hexamers) and the associated transitions is highly complex. However, the authors have used Hidden Markov Model (HMM), which seems to have worked extremely well (being free from systematic error) in extracting the parameters such as oligomeric states, dwell times and transition densities. I think that the authors have done a remarkable job in deciphering such a complex system. The approach used here powerful and quite general, hence it is potentially applicable to many other systems.

Minor concern:

i) Fig 1D is wrongly labeled as E.

Kanchan Garai

Reviewer #2 (Remarks to the Author):

The authors have extensively studied the role of hexamerization of insulin via assembly pathway rerouting revealed by single particle studies". The authors have performed a varied set of experiments to prove that insulin self-assembly can be rerouted by factors such as concentration additives and formulations. Also, the authors have stated the importance of oligomerization and hexamerization. I would appreciate the group for digging out the thermodynamic parameters. I have very few concerns as given below.

To get a more clear idea of the hexamerization the authors must add additional biophysical characterization using DLS, Size exclusion chromatography, and native PAGE data revealing the important transition states as observed.

It is well known and several studies suggest that insulin oligomers are toxic. So if the hexamerization is routed via oligomers what will be the self-assembly process? will the molecules tend to interact with each other by a self-association mechanism or templating mechanism?

I believe it will be more informative for the readers if the authors could give some glimpse on the toxicity profile of each complex monomer -hexamer and hexamer to monomer with and with our additives.

Response Letter

Reviewer #1 (Remarks to the Author):

In this article Bohr et al have utilized single molecule imaging to characterize the oligomerization behaviour of atto655 labelled human insulin (HI655). The authors have shown that labeling of HI by atto655 does not alter the oligomerization equilibrium. Hence, it is possibly safe to assume that oligomerization of HI655 is quite similar to that of unlabeled HI. They have used TIRFM to the record photon count (time) traces of single molecules of monomers and oligomers of HI655. The oligomeric status is determined from the magnitude of the photon counts per second. Recording of time trace has allowed the authors to follow the dwell time of each oligomeric species and the transition events of each oligomer to another type of oligomer. Use of wide field detection allows recording of thousands of such events enabling them to extract the equilibrium and the kinetic parameters of the oligomerization process. One might assume that labeling of HI by atto655 could affect its oligomerization properties. However, the authors have verified that HI655 behaves similarly as HI. One might also think that HI immobilized on the glass would behave differently from the HI in the solution. However, authors have conducted ensemble experiments to verify that the results are consistent with the equilibrium constants obtained from the single molecule experiments. Typically, analysis of the photon count traces involving six different species (monomers to hexamers) and the associated transitions is highly complex. However, the authors have used Hidden Markov Model (HMM), which seems to have worked extremely well (being free from systematic error) in extracting the parameters such as oligomeric states, dwell times and transition densities. I think that the authors have done a remarkable job in deciphering such a complex system. The approach used here powerful and quite general, hence it is potentially applicable to many other systems.

Minor concern:

i) Fig 1D is wrongly labeled as E.

Kanchan Garai

Response

We are very grateful for the reviewer's positive feedback for acknowledging that "the authors have done a remarkable job" and that "the methodology is powerful and quite general, hence it is potentially applicable to many other systems". We are thankful for identifying the wrong labeling of Fig 1D, it is rectified in the revised version.

Reviewer #2 (Remarks to the Author):

The authors have extensively studied the role of hexamerization of insulin via assembly pathway rerouting revealed by single particle studies". The authors have performed a varied set of experiments to prove that insulin self-assembly can be rerouted by factors such as concentration additives and formulations. Also, the authors have stated the importance of oligomerization and hexamerization. I would appreciate the group for digging out the thermodynamic parameters. I have very few concerns as given below.

Comment

To get a more clear idea of the hexamerization the authors must add additional biophysical characterization using DLS, Size exclusion chromatography, and native PAGE data revealing the important transition states as observed.

Response

We agree with the reviewer that biophysical tools are important to cross validate our readouts. We had highlighted in the manuscript (original ref 7,9) that current state of the art insulin hexamerization studies rely on bulk biophysical characterization, like the reviewer asks, and because all these are repeatedly published by many groups for decades, we had not performed them as well. These published method included size exclusion chromatography SEC¹ and Dynamic Light Scattering DLS² are extensively discussed in the cited review³ that addresses the methods and readouts extracting the equilibrium constant. We also had included DLS measurements at two different concentration in our original submission Supplementary Fig. 10. We highlight that none of these methods is able to observe directly the existence and abundance of the intermediates (we believe the reviewer means intermediates when writing "transition states") as they are masked due to averaging the biophysical properties of a large ensemble of unsynchronized molecules. This is the very reason the community relies on fitting the experimental observation with models assuming dimeric or tetrameric additions.

To address fully the comment of the reviewer asking for additional biophysical characterization we performed DLS and SEC that are in general more quantitative than native gels. In detail we :

- a) complemented our existing DLS measurements with DLS at 4 different concentration (from 10-100 μM) in the presence of Zn^{2+} , that is shifting the insulin equilibrium to higher oligomeric states
- b) provided SEC measurements of human insulin and NovoRapid at different concentrations and compared directly the average size of the two form at different concentrations

We are happily reporting that our readouts are consistent with earlier assertions, showing that increasing the insulin concentration resulted in an increase in the average molecular weight and dimension of the particles see Supplementary Fig. 11 and 12 in the revised manuscript. NovoRapid on the other hand, that is engineered to remain monomeric, appears to be unaffected by these changes in concentration. One can appreciate the power of the single molecule readouts in providing the intermediates and their kinetics when compared an average dimension by DLS and SEC.

Changes in the manuscript

We have added these new figures as Supplementary Fig. 11 and Supplementary Fig. 12 and discussed this in the main text in page 7 in section *Quantification of the abundance of oligomeric states and the kinetics of transitions between them and Monomeric assembly and disassembly pathway for HI at nM concentration in the absence of additives.*

We also clarified further that this is the first direct and detailed extraction of rates constants for all intermediates, in section *Monomeric assembly and disassembly pathway for HI at nM concentration in the absence of additives* specifically discussing NovoRapid that "*While averaging techniques are unable to extract each individual intermediate the single particle results here revealed , the association rates involving dimer addition or higher-order oligomers and that they were reduce supporting and extending current understanding*"

We have added a section in Methods describing the DLS data.

Comment

It is well known and several studies suggest that insulin oligomers are toxic. So if the hexamerization is routed via oligomers what will be the self-assembly process? will the molecules tend to interact with each other by a self-association mechanism or templating mechanism? I believe it will be more informative for the readers if the authors could give some glimpse on the toxicity profile of each complex monomer -hexamer and hexamer to monomer with and with our additives.

Response

We agree with the reviewer that insulin aggregation and the formation of higher order aggregates are potential toxic. They have been found deposited at diabetic patients' insulin injection sites⁴, and in the brain tissues in connection with the onset and progression of Alzheimer's disease⁵. As a matter of fact, we have developed a new super resolution method to observed directly the abundance, heterogeneity and kinetics of formation of diverse insulin aggregates, and how their morphological integrity is dependent on environmental conditions and currently (in review) are examining the immune response of them.

We had not included this in the original submission as aggregation and oligomerization are two distinct mechanisms. To fully address the comment of the reviewer we have clarified this further in the manuscript

Changes in the manuscript

We have included a section in page 5 discussing the self assembled hexamers reported here are distinct from the toxic insulin aggregates and that we have recorded the aggregates using super resolution.

We finally performed minor generic changes in the manuscript to rectify spelling phrasing or editing mistakes and renumbering the SI figures.

References

- 1 Brems, D. N. *et al.* Altering the association properties of insulin by amino acid replacement. *Protein Eng.* **5**, 527-533, doi:10.1093/protein/5.6.527 (1992).
- 2 Kadima, W. *et al.* The influence of ionic strength and pH on the aggregation properties of zinc-free insulin studied by static and dynamic laser light scattering. *Biopolymers* **33**, 1643-1657, doi:<https://doi.org/10.1002/bip.360331103> (1993).

- 3 Søbørg, T., Rasmussen, C. H., Mosekilde, E. & Colding-Jørgensen, M. Absorption kinetics of insulin after subcutaneous administration. *Eur. J. Pharm. Sci.* **36**, 78-90, doi:10.1016/j.ejps.2008.10.018 (2009).
- 4 Yumlu, S., Barany, R., Eriksson, M. & Röcken, C. Localized insulin-derived amyloidosis in patients with diabetes mellitus: a case report. *Hum. Pathol.* **40**, 1655-1660, doi:<https://doi.org/10.1016/j.humpath.2009.02.019> (2009).
- 5 House, E., Jones, K. & Exley, C. Spherulites in Human Brain Tissue are Composed of Beta Sheets of Amyloid and Resemble Senile Plaques. *J. Alzheimers Dis.* **25**, 43-46, doi:10.3233/JAD-2011-110071 (2011).